# Prevalence of multimorbidity and its relationship with socioeconomic status among Chinese older adults over time

Qin Liu[ORCID][1]*, Jiehua Lu[2]

1 Research Institute of Social Development, Southwestern University of Finance and Economics, Chengdu, China, 2 Department of Sociology, Peking University, Beijing, China

* demoqin@163.com

## Abstract

Previous studies linking socioeconomic status (SES) to chronic diseases tended to focus on a single disease. As people age, they are more likely to suffer from multiple coexisting chronic conditions, known as multimorbidity. The study of multimorbidity is one of the key links to understanding the impact of population ageing from a comprehensive perspective. This study used four waves of cross-sectional data from the China Health and Retirement Longitudinal Study (CHARLS) from 2011 to 2018 to explore the prevalence of multimorbidity and its relationship with socioeconomic status among older adults in China over time. Participants aged 60 and older were selected for analysis. Both the Logistic Regression Model and the Negative Binomial Regression Model were adopted to examine the relationship between socioeconomic status and multimorbidity. The results showed that the prevalence of multimorbidity among older adults in China demonstrated an increasing trend over the years, from 46.16% in 2011 to 57.50% in 2018. A significant association was detected between socioeconomic status and multimorbidity among older adults, which was manifested as the higher the socioeconomic status, the greater the likelihood of being multimorbid. However, the relationship between the two has been changing over time, with the influence of SES on multimorbidity gradually disappearing and then reappearing in the opposite direction. Multimorbidity has become a critical health issue that should not be ignored for older adults in China, and the relationship between socioeconomic status and multimorbidity may be changing over time, which needs to be further explored with data over a longer period of time.

## Introduction

Nowadays, most health problems in older adults are caused by chronic diseases [1]. In the past two decades, China has witnessed a rapid shift in the disease spectrum, and the burden of chronic diseases among older adults has taken the first place

**Data availability statement:** The data analyzed in this study are publicly available from the China Health and Retirement Longitudinal Study (CHARLS) dataset, https://charls.charls-data.com/pages/data/111/zh-cn.html.

**Funding:** The author(s) received no specific funding for this work.

**Competing interests:** The authors have declared that no competing interests exist.

among various diseases [2]. Active response to chronic diseases has become an important issue of health ageing in China and the world in the 21st century.

As people age, they are more likely to experience multimorbidity, that is, the coexistence of two or more chronic conditions in one person at the same time [1,3]. A review synthetizing studies from seven high-income countries found that more than 50 percent of older adults suffered from multimorbidity, with the prevalence rising sharply among the oldest old [4]. The integrated impact of multimorbidity on functional capacity, quality of life and mortality risk is thought to be significantly greater than the sum of the separate effects of these diseases [4,5], resulting in an enormous burden of disease for individuals, families and society [6,7]. Therefore, the study of multimorbidity is one of the key links to understanding the impact of population ageing from a comprehensive perspective [8].

Socioeconomic status (SES) has been considered to be one of the fundamental causes of disease in recent research [9]. Previous studies linking socioeconomic status to chronic diseases tended to focus on a single disease [10]. Nevertheless, the impact of social factors on health is not specific to any one disease, but general and cumulative. Therefore, analyses that focus on a single outcome may underestimate the impact of social factors on health or disease. As multimorbidity has become a common health issue, the research perspective on the social determinants of disease should shift from a single disease to multimorbidity. Many studies have examined the relationship between SES and multimorbidity, yet findings were still controversial. Some results showed that higher SES (including household/personal income, education level, etc.) was associated with a higher risk of suffering from multimorbidity in older adults [11–17]. However, other studies found that SES exerted a negative impact on the occurrence of multimorbidity, that is, people with lower education level, lower income, lower occupational status and poorer living conditions were more likely to be multimorbid [18–22].

It is noteworthy that most studies came from developed countries, reflecting the characteristics of multimorbidity in Western populations, while few studies systematically explored the relationship between the two in the context of developing countries, such as China. In addition, studies mainly used cross-sectional data in a single year and have not yet analyzed in depth whether the relationship between multimorbidity and SES has changed or is changing with the development of society, economy and population. Therefore, this study used four waves of data from the China Health and Retirement Longitudinal Study (CHARLS) to explore the prevalence of multimorbidity and its relationship with SES among older adults in China from 2011 to 2018.

## Methods

### Data source and processing

This study used four waves of cross-sectional data from the China Health and Retirement Longitudinal Study (CHARLS) between 2011 and 2018 to explore the trend of the relationship between multimorbidity and SES over time, which was available through the website of the National Development Institute of Peking

University [23]. CHARLS is a large-scale interdisciplinary survey project hosted by the National School of Development of Peking University and jointly carried out by the Institute of Social Science Survey and the Communist Youth League Committee of Peking University. The purpose of this survey was to collect a set of high-quality microdata representing households and individuals aged 45 and over, to facilitate an in-depth exploration of the population ageing issue in China. The national baseline survey was carried out in 150 counties and 450 communities (villages) in 28 provinces of China in 2011, followed by three waves of surveys in 2013, 2015 and 2018 respectively; as of 2018, its sample had covered 19,000 respondents from a total of 12,400 households [24]. All participants provided written informed consent at the time of participation and the Biomedical Ethics Review Committee of Peking University approved the CHARLS data collection (IRB00001052–11015).

Participants aged 60 and above who had participated in the survey between 2011 and 2018 were selected for analysis. To make each wave of survey data representative of the national older population, this study adjusted three types of weights, namely the basic sampling weight, the post-stratification weight and the proportional correction weight. Multiple imputation method was adopted to impute the missing values for the data in the third wave (2015) to avoid large losses, as missing values were found in about 20% of the cases in this wave. Because the proportions of cases with missing values in the other three waves (2011, 2013 and 2018) were within a reasonable interval, the listwise deletion method was used directly.

## Measures

**Multimorbidity.** In CHARLS, participants were asked whether they had been diagnosed by a physician with some of 14 listed chronic conditions, including hypertension, dyslipidemia, diabetes or hyperglycemia, cancer or malignant tumor, chronic lung disease, liver disease, heart disease, stroke, kidney disease, digestive disease, emotional or psychiatric problems, memory-related disease, arthritis or rheumatism, and asthma.

Multimorbidity is defined as the coexistence of two or more chronic conditions in one person at the same time [1]. We created a dichotomous variable to indicate the presence of multimorbidity. Participants were considered multimorbid if they reported two or more chronic conditions, assigned a value of 1. In addition, the number of chronic conditions was used as an alternative measure of multimorbidity to ensure the reliability of the results.

**Socioeconomic status.** Education, occupation and income are the three most commonly used indicators to measure socioeconomic status (SES) [25,26]. Since this study was conducted with older people aged 60 and over, it was necessary to consider whether the measures of SES were appropriate for the characteristics of older adults, as well as endogenous problems caused by reverse causation among variables. In this study, two indicators, the education level and the level of annual household income per capita, were used to measure SES of older adults. The education level of older adults was classified into three levels: below primary school, primary school, secondary school and above. In this study, the four aspects of household income in CHARLS were summed and divided by the number of people living in the household to obtain the annual household income per capita. Then, according to its quartile distribution, this kind of income was converted into four levels and was classified as: low income, lower middle income, upper middle income, and high income.

**Covariates.** Covariates were demographic characteristics and medical conditions. Demographic characteristics included age, gender, residence and marital status. Among them, age was a continuous variable; residence was classified as "urban area" and "rural area"; marital status was classified as "having a spouse" and "having no spouse". Variables of medical conditions included medical examination and medical insurance, both of which were treated as dichotomous.

## Statistical analysis

In this study, all analyses were conducted under the condition of data weighting. Percentages, means, standard errors and graphical displays were used for descriptive analyses. The Chi-squared test was used to measure differences in the

prevalence of multimorbidity between variables. Logistic Regression Model was conducted to examine the relationship between multimorbidity (dichotomous) and SES, controlling for demographic and medical factors. Factors associated with multimorbidity were reported as adjusted odds ratios (aORs) and 95% confidence intervals (95% CIs).

To ensure the reliability of results, the number of chronic conditions was used as an alternative measure of multimorbidity for robust analysis. Since the number of chronic conditions was a counting variable, Negative Binomial Regression Model was conducted for analysis, and the results were reported as adjusted incidence-rate ratios (aIRRs) and 95% confidence intervals (95% CIs).

Since the prevalence of multimorbidity presented an "inverted U-shaped" curve with increasing age, the quadratic form of age was included in models to better fit these models. To facilitate the interpretation of results and prevention of multi-collinearity between variables, age was centered at 60 years old. Relationships were regarded as statistically significant if P-values were less than or equal to 0.05. All analyses were performed using Stata 16 MP.

## Results

### Sample characteristics

From 2011 to 2018, the average age of older adults was about 69 years old, with an even gender distribution. In terms of SES, the proportion of older adults with an education level of middle school and above has gradually increased over time, indicating that the education level of older adults has improved. Furthermore, the proportion of older adults who had undergone medical examinations has risen through the years, while the proportion of older adults who had medical insurance kept above 90%. More details of the sample characteristics are presented in Table 1.

### Prevalence of multimorbidity among older adults

As shown in Table 1, the prevalence of multimorbidity among older adults reached 46.16% in 2011 and has been on the rise over time, exceeding 50% after 2013 and even approaching 60% in 2018. The average number of chronic conditions was 1.66 in 2011, which increased to 2.23 in 2018, gradually showing a status quo of "being multimorbid per capita".

**Table 1. Demographic characteristics, SES and multimorbidity of older adults from 2011 to 2018.**

| Variables | 2011 | 2013 | 2015 | 2018 |
|---|---|---|---|---|
| | (N = 7049) | (N = 8284) | (N = 9573) | (N = 10214) |
| Being multimorbid, % | 46.16 | 50.31 | 52.61 | 57.50 |
| Number of chronic conditions, M(SE) | 1.66(0.04) | 1.82(0.04) | 1.91(0.03) | 2.23(0.03) |
| Age, M(SE)[a] | 69.03(0.14) | 69.37(0.13) | 69.12(0.15) | 69.22(0.13) |
| Male, % | 49.08 | 49.43 | 48.79 | 48.64 |
| Having a spouse, % | 75.27 | 75.76 | 77.33 | 78.16 |
| Urban area, % | 42.72 | 43.42 | 49.84 | 49.89 |
| Education level (Below primary school, %) | 36.99 | 35.85 | 32.59 | 28.13 |
| Primary school | 42.85 | 42.27 | 43.25 | 41.97 |
| Secondary school and above | 20.16 | 21.88 | 24.61 | 29.90 |
| Household income level (Low income, %) | 24.15 | 25.45 | 33.61 | 21.88 |
| Lower middle income | 22.22 | 22.57 | 16.75 | 21.57 |
| Upper middle income | 24.34 | 24.08 | 22.67 | 23.67 |
| High income | 29.29 | 27.90 | 26.97 | 32.88 |
| Having medical examination, % | 59.91 | 79.58 | 85.91 | 90.64 |
| Having medical insurance, % | 93.26 | 95.82 | 89.94 | 97.02 |

[a]Abbreviations: SE, Standard Error.

Table 2 shows the trend in the prevalence of multimorbidity among older adults with different demographic, socioeconomic and medical characteristics from 2011 to 2018. The prevalence of multimorbidity among older adults in different age groups showed an increasing trend year by year, especially among the middle-aged (70–79 years old) and advanced-aged (80 years old and above) older adults, whose prevalence of multimorbidity showed a sharp rise, reaching around 70% in 2018. Over time, the prevalence of multimorbidity among older adults who were females and living in urban areas continued to be statistically higher than those who were males and living in rural areas. From the aspect of medical characteristics, older adults who have undergone medical examinations showed a higher prevalence of multimorbidity, which may be related to the higher detection rate of chronic diseases.

With respect to SES, the prevalence of multimorbidity among older adults of different SES basically showed an increasing trend over time, but changes seem to have taken place. Specifically, Table 2 and Fig 1 show that, from 2011 to 2015, the higher the education level of older adults, the higher the prevalence of multimorbidity. In 2018, however, the overall trend was reversed, showing that the higher the education level, the lower the prevalence of multimorbidity. In addition, as time went on, the prevalence curves of multimorbidity at different household income levels crossed (Fig 2), reflecting the reversal of the relative position of the prevalence of multimorbidity among older adults at different household income levels. These results suggest that the relationship between SES and multimorbidity has gradually changed among Chinese older adults, which was further explored by the regression analysis in this study.

### Relationship between multimorbidity and SES over time

Four waves of cross-sectional data from 2011 to 2018 were used to explore the trend in the relationship between multimorbidity and SES among older adults over time, with results shown in Table 3. After controlling for covariates, the education level remained a significant impact on multimorbidity from 2011 to 2015. Specifically, compared with older adults with an education level below primary school, older adults with an education level of primary school and secondary school or above were more likely to suffer from multimorbidity, reflecting that the higher the education level, the higher the likelihood of being multimorbid in older adults. However, in 2018, this relationship was no longer significant; higher levels of education did not correspond to higher prevalence rates of multimorbidity. In addition, compared with low household income, the adjusted odds ratios of middle and high household income were close to or less than 1.0 but not statistically significant in 2015 and 2018, which might predict a reversal in the direction of SES's influence on multimorbidity.

To ensure the robustness of results, the Negative Binomial Regression Model was conducted by changing the dependent variable to another measure of multimorbidity (i.e., the number of chronic conditions). The results, shown in Table 4, are basically consistent with those in Table 3, demonstrating the robustness of our findings. Specifically, from 2011 to 2018, the education level remained a significant impact on the number of chronic conditions among older adults. Older adults with an education level of primary school and above had more chronic diseases than those with an education level below primary school consistently. Furthermore, compared with those with low household income, older adults with high or upper middle household income had more chronic diseases in 2011 and 2013. While in 2015, compared with older adults with low household income, older adults with high household income suffered from fewer chronic diseases, indicating a reversal of the influence direction.

In general, the results in Table 3 and Table 4 basically confirmed the trend of the relationship between multimorbidity and SES in the descriptive analysis (see Table 2, Figs 1 and 2). The relationship between multimorbidity and SES among older adults in China has been changing over time, with the influence of SES on multimorbidity gradually disappearing and subsequently reappearing in a reversed direction. However, the trend was currently unstable.

## Discussion

### The positive relationship between multimorbidity and SES: three explanatory perspectives

Socioeconomic status reflects the social environment in which a person lives throughout life and is a strong predictor of an individual's health status [27]. Many previous studies have shown that people with higher SES were more likely to have

**Table 2. Prevalence of multimorbidity by demographic, socioeconomic and medical characteristics from 2011 to 2018.**

| Variables (%) | 2011 (N = 7049) | 2013 (N = 8284) | 2015 (N = 9573) | 2018 (N = 10214) |
|---|---|---|---|---|
| **Age** | | | | |
| 60-69 | 45.77 | 49.16 | 49.66 | 48.91 |
| 70-79 | 48.16 | 52.02 | 56.32 | 70.74 |
| ≥80 | 41.87 | 49.84 | 59.05 | 68.58 |
| P-value[a] | 0.114 | 0.213 | ——[b] | <0.001 |
| **Gender** | | | | |
| Male | 44.38 | 47.67 | 50.01 | 54.29 |
| Female | 47.87 | 52.54 | 55.08 | 60.36 |
| P-value | 0.035 | 0.002 | —— | <0.001 |
| **Marital status** | | | | |
| Having a spouse | 46.12 | 49.93 | 51.52 | 55.17 |
| Having no spouse | 46.29 | 51.50 | 56.29 | 65.83 |
| P-value | 0.927 | 0.375 | —— | <0.001 |
| **Residence** | | | | |
| Urban area | 49.54 | 52.24 | 56.09 | 59.28 |
| Rural area | 43.64 | 48.52 | 49.15 | 55.55 |
| P-value | 0.005 | 0.024 | —— | 0.008 |
| **Education level** | | | | |
| Below primary school | 43.31 | 48.25 | 51.46 | 59.81 |
| Primary school | 47.46 | 49.63 | 51.76 | 59.17 |
| Secondary school and above | 48.63 | 54.61 | 55.60 | 52.96 |
| P-value | 0.023 | 0.045 | —— | <0.001 |
| **Household income level** | | | | |
| Low income | 43.29 | 47.40 | 55.38 | 60.26 |
| Lower middle income | 45.17 | 49.91 | 53.12 | 58.20 |
| Upper middle income | 44.33 | 52.26 | 51.91 | 55.23 |
| High income | 50.80 | 51.22 | 49.42 | 56.83 |
| P-value | 0.004 | 0.065 | —— | 0.081 |
| **Medical examination** | | | | |
| Having | 51.05 | 52.37 | 54.32 | 58.97 |
| Not having | 38.85 | 42.31 | 42.20 | 43.24 |
| P-value | <0.001 | <0.001 | —— | <0.001 |
| **Medical insurance** | | | | |
| Having | 46.46 | 50.65 | 53.25 | 57.52 |
| Not having | 42.01 | 42.67 | 46.90 | 56.83 |
| P-value | 0.127 | 0.022 | —— | 0.847 |

[a]Differences between the prevalence of multimorbidity within each variable were measured using the Chi-squared test.

[b]Due to the multiple imputation, the data in 2015 cannot be tested by Chi-squared test in Stata.

access to health-related resources, privileges and awareness for better health [28–30]. As a result, lower socioeconomic groups were disadvantaged in health and suffer increased disease and disability [31–33]. However, this study found that there was mainly a significant positive relationship between SES and multimorbidity among older adults over time, that is, the higher the SES, the greater the likelihood of being multimorbid or suffering from more chronic conditions. This positive

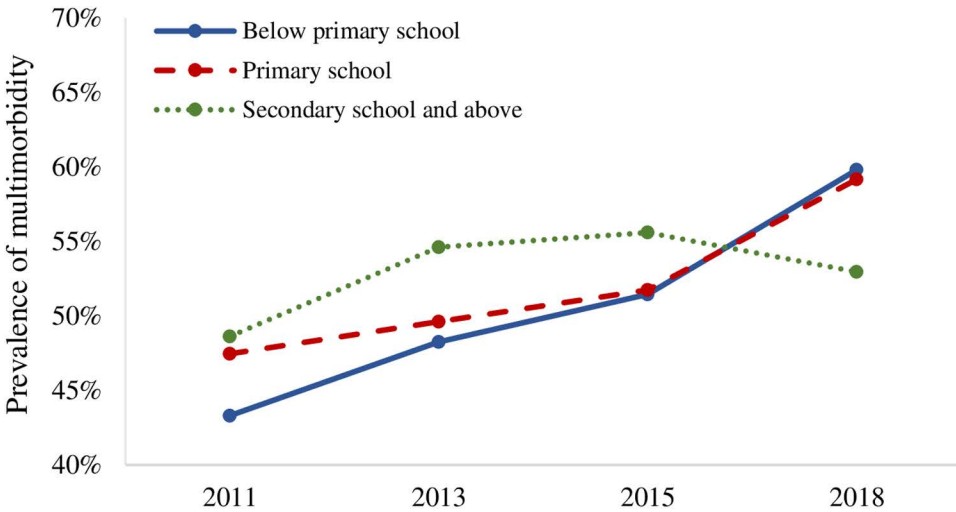

**Fig 1. Prevalence of multimorbidity among older adults with different levels of education.**

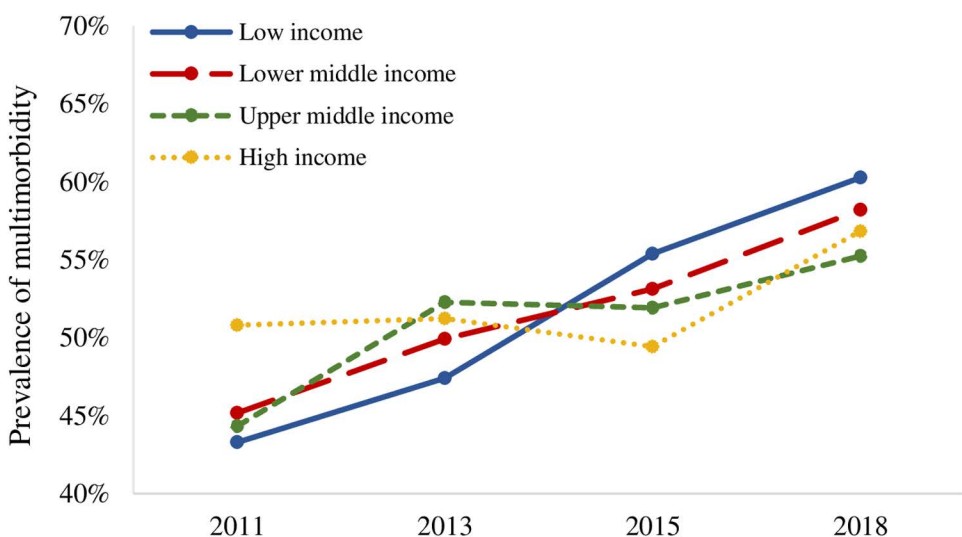

**Fig 2. Prevalence of multimorbidity among older adults with different levels of household income.**

relationship between SES and multimorbidity can be explained from three perspectives, namely, the high prevalence of lifestyle-related diseases, the equity of access to medical services, and the death selectivity of chronic diseases.

The first explanation is the high prevalence of lifestyle-related diseases. Studies have verified that some kinds of chronic diseases (e.g., high blood pressure, heart diseases, metabolic diseases, etc.) were highly associated with factors such as unhealthy lifestyles or stressful states, which were more prevalent among people with higher SES [34–38], resulting in a higher risk of being multimorbid.

Second, due to the "need to be diagnosed" characteristics of diseases, another perspective to explain findings of this study was the equity of access to medical services for older adults with different SES. Medical expenses have risen rapidly in China over the past few decades [39], while people's decisions to utilize medical services were strongly

**Table 3.** Results of the Logistic Regression Model of the relationship between being multimorbid[a] and SES from 2011 to 2018.

| | 2011 (N = 7049) | 2013 (N = 8284) | 2015 (N = 9573) | 2018 (N = 10214) |
|---|---|---|---|---|
| **Education level** (Below primary school = 0) | | | | |
| Primary school | 1.242** | 1.175* | 1.107 | 1.088 |
| | (1.055, 1.461)[b] | (1.035, 1.333) | (0.932, 1.316) | (0.958, 1.235) |
| Secondary school and above | 1.158 | 1.408** | 1.261* | 1.106 |
| | (0.932, 1.438) | (0.123, 1.765) | (1.004, 1.584) | (0.933, 1.312) |
| **Household income level** (Low income = 0) | | | | |
| Lower middle income | 1.094 | 1.137 | 1.055 | 1.039 |
| | (0.927, 1.291) | (0.974, 1.327) | (0.905, 1.231) | (0.886, 1.219) |
| Upper middle income | 1.013 | 1.206* | 0.967 | 1.014 |
| | (0.851, 1.207) | (1.023, 1.422) | (0.823, 1.135) | (0.867, 1.186) |
| High income | 1.186 | 1.040 | 0.862 | 0.930 |
| | (0.967, 1.455) | (0.856, 1.262) | (0.729, 1.019) | (0.772, 1.120) |
| **Demographic characteristics** | Controlled | Controlled | Controlled | Controlled |
| **Medical conditions** | Controlled | Controlled | Controlled | Controlled |

[a]Dependent variable: whether to be multimorbid (1 = Yes; 0 = No).

[b]Results were reported as adjusted odds ratio (aOR) and 95% confidence interval (95% CI); *** p <0. 001, ** p < 0. 01, * p < 0. 05.

**Table 4.** Results of the Negative Binomial Regression Model of the relationship between the number of chronic conditions[a] and SES from 2011 to 2018.

| | 2011 (N = 7049) | 2013 (N = 8284) | 2015 (N = 9573) | 2018 (N = 10214) |
|---|---|---|---|---|
| **Education level** (Below primary school = 0) | | | | |
| Primary school | 1.141*** | 1.079** | 1.057 | 1.055* |
| | (1.059, 1.229)[b] | (1.018, 1.143) | (0.990, 1.129) | (1.004, 1.109) |
| Secondary school and above | 1.134* | 1.192** | 1.130* | 1.045 |
| | (1.029, 1.250) | (1.044, 1.360) | (1.023, 1.239) | (0.978, 1.117) |
| **Household income level** (Low income = 0) | | | | |
| Lower middle income | 1.063 | 1.064 | 1.020 | 1.026 |
| | (0.986, 1.147) | (0.994, 1.138) | (0.951, 1.094) | (0.972, 1.082) |
| Upper middle income | 1.006 | 1.154** | 1.016 | 1.022 |
| | (0.930, 1.089) | (1.051, 1.268) | (0.951, 1.085) | (0.968, 1.079) |
| High income | 1.110* | 1.043 | 0.920* | 0.990 |
| | (1.008, 1.221) | (0.938, 1.160) | (0.854, 0.991) | (0.926, 1.057) |
| **Demographic characteristics** | Controlled | Controlled | Controlled | Controlled |
| **Medical conditions** | Controlled | Controlled | Controlled | Controlled |

[a]Dependent variable: the number of chronic conditions.

[b]Results were reported as adjusted incidence-rate ratio (aIRR) and 95% confidence interval (95% CI); *** p < 0. 001, ** p < 0. 01, * p < 0. 05.

influenced by their economic affordability. The poorest people often did not have sufficient money to pay for necessary health services, resulting in delay or abandonment of medical care and under-diagnosis of chronic diseases [40]. In addition, people with lower education levels were less likely to perceive diseases by themselves due to lack of ability, increasing the possibility of underreporting diseases in surveys [41]. In contrast, older adults with higher SES were more likely to undergo regular medical examinations or own medical insurance, thereby detecting and reporting more chronic diseases.

Finally, the death selectivity of chronic diseases also might exaggerate the positive association between SES and multimorbidity. According to the "expansion of disease hypothesis", as medical technology and lifestyles improved, advances in life-saving technologies outweighed advances in health-promoting technologies, which led to surviving population living longer with poorer health [42,43]. Older adults with higher SES may survive longer with multimorbidity due to early detection, timely treatment and long-term care; while older adults with lower SES may die from illness earlier due to their poorer health status and less access to better health services, resulting in older adults with higher SES appearing to be in poorer health [44]. As the results of this study showed, older adults with higher SES performed a higher prevalence of multimorbidity and suffered from more chronic diseases.

### Trends in the relationship between multimorbidity and SES

Previous studies on multimorbidity in China, South Africa, Bangladesh, Brazil and other developing countries showed that people with higher SES were more likely to be multimorbid [11–15,17], while most studies in developed countries have found that people in disadvantaged SES were more likely to develop multimorbidity [4,20,22,45–51]. Some researchers believed that the impact of SES on health might vary with different stages of social development in countries [52]. The results of this study suggest that China may be experiencing a transition in the relationship between SES and multimorbidity over time, and may gradually exhibit a more significant and stable relationship in the future.

In countries at different stages of development, differences in the relationship between SES and chronic diseases may be due to changes in lifestyle and disease caused by socioeconomic development. To be specific, in some low- and middle-income countries, people's lifestyles has been undergoing the early process of "westernization" during the transitional phase of socioeconomic development [34]. People with higher SES were the first to practice unhealthy lifestyles, such as eating high-calorie or processed foods and consuming tobacco and alcohol; at the same time, with the development of science and technology, people with higher SES were more likely to work long hours in their seats or in front of computer screens, further reducing their daily activities and leading to a higher risk of suffering from chronic diseases [53,54]. Meanwhile, developed countries have entered a new stage of development, and people with higher SES were starting to pursue healthy diets, good living habits and regular exercises; while people with lower SES, due to high unemployment rate and low income, often ate unhealthy fast food and smoke or drink more, resulting in a higher risk of suffering from chronic diseases [55,56]. Therefore, developing countries may be undergoing the lifestyle and disease transformation experienced by developed countries because of socioeconomic development, leading to a shift in the relationship between SES and chronic diseases.

In addition, as previously stated, due to the "need to be diagnosed" characteristics of diseases, the equity of access to medical services for older adults may affect the relationship between SES and multimorbidity. At present, China has actively promoted the development of diagnosis and treatment in basic medical institutions, and has established a large number of basic medical institutions in rural areas, greatly promoting medical equity. People with lower SES could benefit from medical services more conveniently. Furthermore, the development of the New Rural Cooperative Medical System provided partial financing guarantee for medical expenses, which improved the enthusiasm for medical treatment among people with lower SES [57,58]. Timely and sustained medical care may lead to higher rates of disease detection and increase the likelihood of living with disease among people with lower SES, leading to further changes in the relationship between SES and multimorbidity.

## Strengths and limitations

This study has the following strengths. First, studies on SES and multimorbidity were mainly conducted in European and American countries, reflecting the characteristics of western populations. In this study, national data was used to systematically explore the relationship between SES and multimorbidity and its development in the Chinese context. Second, most existing studies used cross-sectional data in a single year. This study used four waves of data from 2011 to 2018 to explore the prevalence of multimorbidity and its relationship with SES among older adults in China over time, which is helpful to dynamically understand the trend of the relationship between SES and multimorbidity over a long period.

However, there are also some shortcomings in this study. First, there was a limited time span to explore the trend of multimorbidity, which is only 8 years. Chronic diseases are persistent, and it may take a considerable time to observe the transition process. Therefore, the change in the relationship between SES and multimorbidity found in this study might lack stability. We believe that there may gradually show a more significant and stable change, similar to that in developed countries, which requires data support over a longer period of time. Second, the problem of sample selectivity. Whether individuals have a particular disease cannot be judged by themselves, but must be diagnosed by some groups with relevant qualifications (such as doctors). Even if people do suffer from a chronic disease, they may self-report that they do not have this disease for lacking a doctor's diagnosis. Therefore, due to under-diagnosis of diseases, the use of self-reported chronic disease as a method of collecting disease information in surveys may lead to sample selectivity bias [59–61]. This study attempted to control for sample selectivity bias by including variables relevant to medical conditions (namely, medical examinations and medical insurances) in the regression models, but it may not be able to completely solve this problem. Third, limitations in the collection of information and types of chronic diseases. In in the CHALRS survey, participants self-reported whether or not they had a chronic condition, so this result may be affected by recall bias. And the types of chronic diseases in CHARLS were limited to 14 categories, with more types being overlooked, which may have missed other types and underestimated the prevalence of multimorbidity among Chinese older adults. In addition, the use of diseases such as kidney disease, digestive diseases and psychiatric diseases was non-specific, resulting in the loss of important information. However, these limitations cannot be addressed in this study due to data limitations. Further research will be conducted in the future to address these limitations as data allow.

## Future research directions and policy recommendations

Multimorbidity has become a critical health issue that cannot be ignored for older adults in China and around the world. With the continuous updating of survey data, the time span of research on multimorbidity is expected to be further extended. In addition, as society evolves, chronic diseases are becoming more prevalent in younger age groups. Although older adults were more likely to suffer from multimorbidity, studies have found that the prevalence of multimorbidity in young and middle-aged populations were not optimistic, and their multimorbidity showed different patterns and impacts from those of older adults [12,62,63], implying that more attention should be paid to future research on multimorbidity in young and middle-aged populations.

In addition, this study also proposed the following three policy recommendations to reduce the high prevalence of multimorbidity among older people in China. First, improving the socioeconomic status of older people, particularly through pension and social security systems, to ensure that they have sufficient financial resources to maintain a healthy lifestyle and access health care. Second, the coverage of primary health care services should be expanded to ensure that older people have access to timely and convenient medical services. Through regular physical examination and health screening, early detection and intervention of chronic diseases can be achieved, as well as a reduction in the incidence of multimorbidity. Finally, the government should launch health education campaigns to promote healthy lifestyles and raise health awareness among older adults. For example, community health service centers can organize regular health lectures and adopt social media to promote health knowledge for older people.

## Conclusions

To conclude, this study found that the prevalence of multimorbidity among Chinese people aged 60 and above was rising over time. Multimorbidity has become a critical health issue that cannot be ignored in promoting healthy ageing in China. Additionally, we found a significant positive association between SES and multimorbidity among older adults, that is, the higher the SES, the greater the likelihood of being multimorbid or suffering from more chronic conditions; however, the relationship between the two has been changing over time, with the influence of SES on multimorbidity gradually disappearing and subsequently reappearing in a reversed direction. Longer-term data are needed to further explore the evolution of the relationship between SES and multimorbidity in the future. And more attention should also be paid to multimorbidity in young and middle-aged populations.

## Acknowledgments

The acknowledgement is to the participants in the China Health and Retirement Longitudinal Study and the members of the survey teams. We are grateful for the data provided by the National School of Development of Peking University, Beijing, China.

## Author contributions

**Conceptualization:** Qin Liu.

**Data curation:** Qin Liu.

**Formal analysis:** Qin Liu.

**Methodology:** Qin Liu, Jiehua Lu.

**Software:** Qin Liu.

**Supervision:** Jiehua Lu.

**Validation:** Qin Liu, Jiehua Lu.

**Writing – original draft:** Qin Liu.

**Writing – review & editing:** Qin Liu, Jiehua Lu.

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
