## [Decision Letter · Decision Letter 0]

29 Jan 2025

PONE-D-24-53720Prevalence of multimorbidity and its relationship with socioeconomic status among Chinese older adults over time: Evidence from CHARLS 2011-2018PLOS ONE

Dear Dr. Liu,

Thank you for submitting your manuscript to PLOS ONE. After careful consideration, we feel that it has merit but does not fully meet PLOS ONE’s publication criteria as it currently stands. Therefore, we invite you to submit a revised version of the manuscript that addresses the points raised during the review process.

**ACADEMIC EDITOR: **Kindly pay attention to each comment from the reviewers and provide a rebuttal for each query.

REVIEWER 1

A well-written article. Some observations are noted as follows:

1. The use of only 14 chronic conditions could have limited the number of chronic diseases reported and hence underestimated the prevalence of multimorbidity.

2. The use of patient self-report can also be affected by recall bias, especially in the older adults who might have memory problems.

an objective assessment of chronic disease could have given the study more strength.

3. The use of diseases like kidney disease, digestive diseases, and psychiatric diseases might be nonspecific and can result in loss of important information.

REVIEWER 2

1. Title of the study

- The title of the study should be concise. The author is advised to avoid using acronyms in the title and remove the timeframe, as it can be addressed in the methodology section.

2. Abstract

- The first time ‘Socioeconomic Status’ appears in the abstract, it should be followed by its acronym in parentheses (SES).

3. Keywords

- Chronic conditions/diseases could be added as another key word.

4. Methods section

a. Data source and processing

- This sentence in the methods section requires revision for improved grammar and clarity, “To analyze the health problems of ageing in the social and economic transition period, this survey has collected sets of high-quality micro-data representing households and individuals of middle-aged and people aged 45 and above in China”.

- Could the authors clarify why they chose to isolate and analyze data only for individuals aged 60 years and above, despite the CHARLS study including older people starting from age 45? Additionally, it might be helpful to define who qualifies as an older adult in the context of this study.

b. Measures

- Multimorbidity – More detailed information is needed to explain why participants were limited to reporting only whether they had any of the 14 listed conditions.

- Socioeconomic status – This sentence needs to be referenced appropriately “Education, occupation and income are the three most commonly used indicators to measure socioeconomic status (SES)”.

5. Results

- Major results – I suggest the authors use ‘Results’ as the heading for this section. The current heading might imply the existence of ‘Minor Results,’ which could be misleading.

6. Discussion

- Main Findings: This subsection largely reiterates content already presented in the results section. It is recommended that this subsection be removed.

- Discussion: The discussion requires further strengthening. The authors primarily focus on the unidirectional relationship between SES and multimorbidity. For example, they should consider discussing instances where multimorbidity was higher among individuals with lower SES compared to the opposite. What possible factors might contribute to such findings, and how do these align with or differ from existing evidence?

- Recommendations: Beyond suggesting further studies, the authors should provide plausible recommendations for addressing the observed trends in multimorbidity in China.

We look forward to receiving your revised manuscript.

Kind regards,

Godpower C. Michael, MBBS, MHE, FWACP

Academic Editor

PLOS ONE

Journal requirements: When submitting your revision, we need you to address these additional requirements. 1. Please ensure that your manuscript meets PLOS ONE's style requirements, including those for file naming. The PLOS ONE style templates can be found at https://journals.plos.org/plosone/s/file?id=wjVg/PLOSOne_formatting_sample_main_body.pdf and https://journals.plos.org/plosone/s/file?id=ba62/PLOSOne_formatting_sample_title_authors_affiliations.pdf. 2. Please match your authorship list in your manuscript file and in the system. 3. Your ethics statement should only appear in the Methods section of your manuscript. If your ethics statement is written in any section besides the Methods, please delete it from any other section. 

Reviewers' comments:

Reviewer's Responses to Questions

**Comments to the Author**

1. Is the manuscript technically sound, and do the data support the conclusions?

Reviewer #1: Yes

Reviewer #2: Yes

2. Has the statistical analysis been performed appropriately and rigorously?

Reviewer #1: Yes

Reviewer #2: Yes

3. Have the authors made all data underlying the findings in their manuscript fully available?

Reviewer #1: Yes

Reviewer #2: Yes

4. Is the manuscript presented in an intelligible fashion and written in standard English?

Reviewer #1: Yes

Reviewer #2: Yes

5. Review Comments to the Author

Reviewer #1: a well-written article. some observations are noted as

1. The use of only 14 chronic conditions could have limited the number of chronic diseases reported and hence underestimated the prevalence of multimorbidity.

2. The use of patient self-report can also be affected by recall bias, especially in the older adults who might have memory problems.

an objective assessment of chronic disease could have given the study more strength.

3. The use of diseases like kidney disease, digestive diseases, and psychiatric diseases might be nonspecific and can result in loss of important information.

Reviewer #2: 1. Title of the study

- The title of the study should be concise. The author is advised to avoid using acronyms in the title and remove the timeframe, as it can be addressed in the methodology section.

2. Abstract

- The first time ‘Socioeconomic Status’ appears in the abstract, it should be followed by its acronym in parentheses (SES).

3. Keywords

- Chronic conditions/diseases could be added as another key word.

4. Methods section

a. Data source and processing

- This sentence in the methods section requires revision for improved grammar and clarity, “To analyze the health problems of ageing in the social and economic transition period, this survey has collected sets of high-quality micro-data representing households and individuals of middle-aged and people aged 45 and above in China”.

- Could the authors clarify why they chose to isolate and analyze data only for individuals aged 60 years and above, despite the CHARLS study including older people starting from age 45? Additionally, it might be helpful to define who qualifies as an older adult in the context of this study.

b. Measures

- Multimorbidity – More detailed information is needed to explain why participants were limited to reporting only whether they had any of the 14 listed conditions.

- Socioeconomic status – This sentence needs to be referenced appropriately “Education, occupation and income are the three most commonly used indicators to measure socioeconomic status (SES)”.

5. Results

- Major results – I suggest the authors use ‘Results’ as the heading for this section. The current heading might imply the existence of ‘Minor Results,’ which could be misleading.

6. Discussion

- Main Findings: This subsection largely reiterates content already presented in the results section. It is recommended that this subsection be removed.

- Discussion: The discussion requires further strengthening. The authors primarily focus on the unidirectional relationship between SES and multimorbidity. For example, they should consider discussing instances where multimorbidity was higher among individuals with lower SES compared to the opposite. What possible factors might contribute to such findings, and how do these align with or differ from existing evidence?

- Recommendations: Beyond suggesting further studies, the authors should provide plausible recommendations for addressing the observed trends in multimorbidity in China.

6. PLOS authors have the option to publish the peer review history of their article (what does this mean? ). If published, this will include your full peer review and any attached files.

**Do you want your identity to be public for this peer review?** For information about this choice, including consent withdrawal, please see our Privacy Policy .

Reviewer #1: **Yes: ** Zainab Abdulazeez Umar

Reviewer #2: **Yes: ** Duncan Kwaitana

---

## [Author Response · Author response to Decision Letter 1]

18 Feb 2025

Dear Editors and Reviewers,

Thank you for your letter and for the reviewers’ comments concerning our manuscript entitled “Prevalence of multimorbidity and its relationship with socioeconomic status among Chinese older adults over time: Evidence from CHARLS 2011-2018” (ID: PONE-D-24-53720). Those comments are all valuable and very helpful for revising and improving our paper. We have studied comments carefully and have made correction which we hope meet with approval. Revised portion are highlighted in yellow. in the paper. The main corrections in the paper and the responds to the reviewers’ comments are as flowing.

Responds to the reviewers’ comments:

Reviewer #1:

1. Comment: The use of only 14 chronic conditions could have limited the number of chronic diseases reported and hence underestimated the prevalence of multimorbidity.

Response: We are grateful to the reviewer for this valuable comment. However, due to data limitations, this comment cannot be addressed in this study. We have added this limitation to the ‘Strengths and limitations’ section.

2. Comment: The use of patient self-report can also be affected by recall bias, especially in the older adults who might have memory problems. an objective assessment of chronic disease could have given the study more strength.

Response: As with the first comment, due to data limitations, this comment cannot be addressed in this study. We have added this limitation to the ‘Strengths and limitations’ section.

3. Comment: The use of diseases like kidney disease, digestive diseases, and psychiatric diseases might be nonspecific and can result in loss of important information.

Response: As with the first comment, this comment cannot be addressed in this study due to data limitations. We have added this limitation to the ‘Strengths and limitations’ section.

Reviewer #2:

1. Comment: The title of the study should be concise. The author is advised to avoid using acronyms in the title and remove the timeframe, as it can be addressed in the methodology section.

Response: We have modified the original title to “Prevalence of multimorbidity and its relationship with socioeconomic status among Chinese older adults over time” by removing the statement after the comma.

2. Comment: The first time ‘Socioeconomic Status’ appears in the abstract, it should be followed by its acronym in parentheses (SES).

Response: We have added the “(SES)” after the first time “Socioeconomic Status” in the first sentence of the abstract.

3. Comment: Keywords: Chronic conditions/diseases could be added as another key word.

Response: We have added “Chronic diseases” to the list of keywords.

4. Comment: Data source and processing: This sentence in the methods section requires revision for improved grammar and clarity, “To analyze the health problems of ageing in the social and economic transition period, this survey has collected sets of high-quality micro-data representing households and individuals of middle-aged and people aged 45 and above in China”.

Response: We have rewritten the sentence to “The purpose of this survey was to collect a set of high-quality microdata representing households and individuals aged 45 and over, to facilitate an in-depth exploration of the population ageing issue in China.” We believe the new sentence is more concise and easier to understand.

5. Comment: Could the authors clarify why they chose to isolate and analyze data only for individuals aged 60 years and above, despite the CHARLS study including older people starting from age 45? Additionally, it might be helpful to define who qualifies as an older adult in the context of this study.

Response: We are grateful to the reviewer for this valuable comment. According to the international standards provided by WHO, older people usually refer to the population aged 60 and above. In response to this opinion, we have the following two explanations.

Firstly, the research interest. Although CHARLS was conducted on population aged 45 years and older, our research interest was the health issue of older adults, and therefore this study focused on older adults.

Secondly, the research results. According to previous studies, the prevalence of chronic diseases and multimorbidity is significantly higher in older adults than in other age groups. Therefore, by focusing on older adults, this study may yield more significant and meaningful results, and be more likely to observe important information. Including the middle-aged population under 60 years of age in the analysis may affect the power of the analysis due to their lower prevalence of chronic diseases, making it more difficult to observe meaningful results.

6. Comment: Multimorbidity – More detailed information is needed to explain why participants were limited to reporting only whether they had any of the 14 listed conditions.

Response: We thank the reviewer for the valuable comment and have made the following two changes.

Firstly, there is an issue with the wording of the original text which could lead to misunderstanding. In the section of “Measures-Multimorbidity”, We have changed “any of 14 listed chronic conditions” to “some of 14 listed chronic conditions” to indicate that participants could select more than one chronic condition from the list.

Secondly, the limitation that the types of diseases were limited to 14 categories due to data limitations has been added into the ‘Strengths and limitations’ section.

7. Comment: Socioeconomic status – This sentence needs to be referenced appropriately “Education, occupation and income are the three most commonly used indicators to measure socioeconomic status (SES)”.

Response: We have added two references after this sentence in the section of “Measures-Socioeconomic status”.

8. Comment: Results - Major results – I suggest the authors use ‘Results’ as the heading for this section. The current heading might imply the existence of ‘Minor Results,’ which could be misleading.

Response: Considering the reviewer’s suggestion, we have revised the title of “Major results” to “Results”.

9. Comment: Discussion - Main Findings: This subsection largely reiterates content already presented in the results section. It is recommended that this subsection be removed.

Response: It is really true as the reviewer’s suggested that this subsection largely reiterates content already presented in the results section. Therefore, we removed the subsection of “Main Findings”.

10. Comment: Discussion: The discussion requires further strengthening. The authors primarily focus on the unidirectional relationship between SES and multimorbidity. For example, they should consider discussing instances where multimorbidity was higher among individuals with lower SES compared to the opposite. What possible factors might contribute to such findings, and how do these align with or differ from existing evidence?

Response: We are very sorry for our negligence of discussion the opposite side of the results, that is, higher multimorbidity among individuals with lower SES.

Although the discussion of the opposite side (i.e., the negative relationship between SES and multimorbidity) is important for the completeness of the discussion, we do not consider it to be the key point of this paper, nor do we need to add too much discussion here. In addition, in the second subsection of the discussion, there was more discussion of both the positive and negative relationship between SES and multimorbidity in different cultures. Therefore, we have added some relevant but limited discussion of the negative relationship at the beginning of the first subsection of the discussion.

11. Comment: Recommendations: Beyond suggesting further studies, the authors should provide plausible recommendations for addressing the observed trends in multimorbidity in China.

Response: Considering this valuable suggestion, we have added some useful policy recommendations after the suggestions for future research directions.

We tried our best to improve the manuscript and made some changes in the manuscript. These changes will not influence the theme and framework of the paper.

We appreciate for Editors’ and Reviewers’ warm work earnestly, and hope that the correction will meet with approval. Thank you very much for your comments and suggestions.

With best regards.

Sincerely yours,

Qin Liu & Jiehua Lu

Corresponding Email: demoqin@163.com

---

## [Decision Letter · Decision Letter 1]

16 Mar 2025

Prevalence of multimorbidity and its relationship with socioeconomic status among Chinese older adults over time

PONE-D-24-53720R1

Dear Dr. Liu,

We’re pleased to inform you that your manuscript has been judged scientifically suitable for publication and will be formally accepted for publication once it meets all outstanding technical requirements.

Kind regards,

Godpower C. Michael, MBBS, MHE, FWACP

Academic Editor

PLOS ONE

Additional Editor Comments (optional):

Reviewers' comments: nil

Reviewer's Responses to Questions

**Comments to the Author**

1. If the authors have adequately addressed your comments raised in a previous round of review and you feel that this manuscript is now acceptable for publication, you may indicate that here to bypass the “Comments to the Author” section, enter your conflict of interest statement in the “Confidential to Editor” section, and submit your "Accept" recommendation.

Reviewer #1: All comments have been addressed

Reviewer #2: All comments have been addressed

2. Is the manuscript technically sound, and do the data support the conclusions?

Reviewer #1: Yes

Reviewer #2: Yes

3. Has the statistical analysis been performed appropriately and rigorously?

Reviewer #1: Yes

Reviewer #2: N/A

4. Have the authors made all data underlying the findings in their manuscript fully available?

Reviewer #1: No

Reviewer #2: Yes

5. Is the manuscript presented in an intelligible fashion and written in standard English?

Reviewer #1: Yes

Reviewer #2: Yes

6. Review Comments to the Author

Reviewer #1: The author has noted the comments made and has been captured in the limitation section. The discussion needs some improvement, especially to address the relationship of multimorbidity and declined or reduced socioeconomic status.

Reviewer #2: Comments have been successfully addressed the comments and where necessary they have provided a rebuttal

7. PLOS authors have the option to publish the peer review history of their article (what does this mean? ). If published, this will include your full peer review and any attached files.

**Do you want your identity to be public for this peer review?** For information about this choice, including consent withdrawal, please see our Privacy Policy .

Reviewer #1: **Yes: ** Zainab Abdulazeez U.

Reviewer #2: **Yes: ** Duncan Kwaitana

---

## [Editor Report · Acceptance letter]

PONE-D-24-53720R1

PLOS ONE

Dear Dr. Liu,

I'm pleased to inform you that your manuscript has been deemed suitable for publication in PLOS ONE. Congratulations! Your manuscript is now being handed over to our production team.

Kind regards,

on behalf of

Dr. Godpower Chinedu Michael

Academic Editor

PLOS ONE